# Improved Training of Physics-informed Neural Networks using Energy-Based priors: A Study On Electrical Impedance Tomography

**Akarsh Pokkunuru**[*]
apokkunu@uncc.edu

**Amirmohammad Rooshenas**[†]
pedram@uic.edu

**Thilo Strauss**[‡]
thilum@gmx.de

**Anuj Abhishek**[§]
anuj.abhishek@uncc.edu

**Taufiquar Khan**[§]
tkhan13@uncc.edu

## Abstract

Physics-informed neural networks (PINNs) are attracting significant attention for solving partial differential equation (PDE) based inverse problems, including electrical impedance tomography (EIT). EIT is non-linear and especially its inverse problem is highly ill-posed. Therefore, successful training of PINN is extremely sensitive to interplay between different loss terms and hyper-parameters, including the learning rate. In this work, we propose a Bayesian approach through data-driven energy-based model (EBM) as a prior, to improve the overall accuracy and quality of tomographic reconstruction. In particular, the EBM is trained over the possible solutions of the PDEs with different boundary conditions. By imparting such prior onto physics-based training, PINN convergence is expedited by more than ten times faster to the PDE's solution. Evaluation outcome shows that our proposed method is more robust for solving the EIT problem. Our code is available at: https://rooshenasgroup.github.io/eit_ebprior/.

## 1 Introduction

Physics-informed neural networks (PINNs) (Raissi et al., 2019) parameterize the solution of a partial differential equation (PDE) using a neural network and trains the neural network to predict the solution's scalar value for any given point inside the problem's domain by minimizing the residual PDE and associated boundary conditions (BCs). Various other factors such as gradient stiffness (Wang et al., 2021) and complex parameter settings, lack of constraints and regularization (Krishnapriyan et al., 2021), often cause major issues in training PINNs and makes them very sensitive to hyper-parameters and regularization. For example, depending on the PDE, the interplay between BCs residuals and PDEs residuals can result in invalid solutions that mostly satisfy one type of constraints over the others. In general, we believe that some of these problems are attributable to the lack of joint representation in PINNs, as these models are assumed to implicitly learn that two points in a close vicinity possibly have a similar solution. However, this representation is hard-coded in the numerical methods as they update the solution value of each mesh point based on the neighboring mesh points.

In this work, we augment PINNs with a joint representation via a Bayesian approach. We train a data-driven prior over joint solutions of PDEs on the entire domain and boundary points. This prior relates the predictions of PINNs via explicit joint representation and encourage learning a coherent and valid solution. Priors have been studied extensively in statistical approach to solve inverse problems (Kaipio et al., 2000; Ahmad et al., 2019; Abhishek et al., 2022; Strauss & Khan, 2015) and also recently have been used in data-driven approaches as well (Ramzi et al., 2020). More importantly, we focused on the non-linear and ill-posed electrical impedance tomography (EIT)

---

[*]Department of Computer Science, University of North Carolina at Charlotte, NC, USA

[†]Department of Computer Science, University of Illinois Chicago, IL, USA

[‡]Bosch ETAS Research, Germany, EU

[§]Department of Mathematics and Statistics, University of North Carolina at Charlotte, NC, USA

inverse problem and experimentally show that using our data-driven Bayesian approach results in accurate, fast, and more robust training algorithms. We have the following contributions:

- We present various experimental studies which circumvent instability in training PINNs for the EIT problem.
- We introduce a data-driven prior in the form of an EBM for improving the training convergence and accuracy of PINNs.
- We propose a robust training framework for the EIT semi-inverse problem.

## 2 AN INVERSE PROBLEM WITH APPLICATION TO ELECTRICAL IMPEDANCE TOMOGRAPHY (EIT)

The inverse problem that we are interested in is inspired by applications in imaging paradigms such as electrical impedance tomography (EIT) and geophysical imaging for ground water flow. The EIT inverse problem reconstructs the unknown electrical conductivities $\sigma$ of a body $\Omega \subset \mathbb{R}^d$ with $d \in \{2, 3\}$ from measurements of finite electrical potential differences of neighboring surface electrodes. The differential equation

$$-\nabla \cdot \sigma \nabla u = 0 \qquad \text{in } \Omega \tag{1}$$

governs the distribution of electric potential $u$ in the body. Additionally, its accompanying BCs are given as following:

$$\sigma \left( \frac{\partial u}{\partial n} \right) = g \qquad \text{on } \partial\Omega \ \text{ Neumann BC,}$$
$$u = f \qquad \text{on } \partial\Omega \ \text{ Dirichlet BC} \tag{2}$$

An EIT experiment involves applying an electrical current on the surface $\partial\Omega$ of the region $\Omega$ to be imaged, which produces a current density $\sigma \frac{\partial u}{\partial n}|_{\partial\Omega} = g$ (Neumann data) where $n$ is a unit normal vector w.r.t $u$ associated with $\Omega$ at its boundary. The current also induces an electric potential $u$ in the body, whose surface value $u|_{\partial\Omega} = f$ can be measured. Thus by repeating several such experiments in which surface current is given and the corresponding surface voltage (Dirichlet data) is measured, we obtain the information on the Neumann-to-Dirichlet (NtD) operator which can be denoted by:

$$\Lambda_\sigma : g \mapsto f. \tag{3}$$

In the full EIT version, the problem is to reconstruct $\sigma$ from just the surface current and voltage measurements. The estimate of the unknown conductivity $\sigma$ can be reconstructed from a set of EIT experiments (Somersalo et al., 1992; Borcea, 2002; Hanke & Brühl, 2003). In a simplified version that we attempt to solve here using our novel method, we will be interested in recovering $\sigma$ from the measurements of $u$ in the interior of the medium. This problem formulation is a close to the groundwater flow problem, wherein the source term on the right hand side of Eq. 1 is non-zero. It should also be noted, that a similar formulation was studied in Bar & Sochen (2021).

In our training paradigm, we need to simulate $u$ for the interior of the medium for any underlying $\sigma$. We achieve that by first solving the forward problem by training neural network that can predict the value of the function $u(x)$ at any given point $x \in \Omega$, for any given $\sigma$. Once we have access to this pre-trained network that we will call $u$-Net, we will subsequently train another network to predict the value of $\sigma(x)$ for any $x \in \Omega$ from point-wise measurements of a function $u(x)$ that satisfies Eq. 1 and agrees with the Neumann and Dirichlet boundary data given from Eq. 3.

### 2.1 USING PINNS FOR EIT

Following the construction of physics-informed neural networks (PINNs), we can parameterize both $\sigma$ and $u$ using neural networks, called $\sigma$-Net and $u$-Net, respectively, and train them such that the values of $\sigma$-Net and $u$-Net for any provided point in the interior or boundary of the medium respect the PDE in Eq.1 or its boundary conditions[1] – depending on the position of the point. The forward

---

[1]To the best of our knowledge only Bar & Sochen (2021) claim to train the EIT inverse problem using PINNs by jointly training $u$-Net and $\sigma$-Net, but unfortunately, their implementation is not open-source and we were unable to reproduce their results based on the details in the paper. See Appendix A.5 for more details.

problem assumes that $\sigma$ distribution is known everywhere inside the domain $\Omega$. The $u$-Net takes mesh points in Cartesian space $(x, y) \in \mathbb{R}^2$ as input and correspondingly learns to recover correct potential values $u$ at the points inside the domain $\Omega$, given the voltages only at electrodes $u_e$. To train $u$-Net, we utilize Eq. 1 and, 2 to write them in a functional form as follows:

$$\mathcal{L}_{\text{PDE}}^d = \nabla \cdot \left( \sigma_d \, \nabla u_d \right) \quad \forall d \in \Omega \tag{4}$$

$$\mathcal{L}_{\text{BC}}^b = \left[ \sigma_b \frac{\partial u_b}{\partial n_b} \right] \quad \forall b \in \partial \Omega_b, \quad \mathcal{L}_{\text{BC}}^e = \left[ \sigma_e \frac{\partial u_e}{\partial n_e} - g_e \right] \quad \forall e \in \partial \Omega_e. \tag{5}$$

Here, $\mathcal{L}_{\text{PDE}}^d$ governs the potential $u$ in forward and conductivity $\sigma$ distributions in inverse problem respectively. $\mathcal{L}_{\text{BC}}^b$ are the combined BCs where we enforce Neumann BC twice, once on all boundary points $\partial \Omega_b$ except electrodes and separately on electrodes $\partial \Omega_e$. Thus, $\sigma$-Net and $u$-Net represent their respective distributions as parameterized functions $\sigma(x, y)$ and $u(x, y)$. We can then write a common objective function which is used to train both problems as follows:

$$\mathcal{L}_\theta = \frac{\alpha}{\Omega} \sum_{d \in \{\Omega\}} (\mathcal{L}_{\text{PDE}}^d)^2 + \frac{\beta}{M} \sum_{m \in \text{top}_M \mathcal{L}_{\text{PDE}}} |\mathcal{L}_{\text{PDE}}| + \frac{\gamma}{|\partial \Omega_b|} \sum_{b \in \partial \Omega_b} |\mathcal{L}_{\text{BC}}^b| + \frac{\delta}{|\partial \Omega_e|} \sum_{e \in \partial \Omega_e} \mathcal{L}_{\text{BC}}^e, \tag{6}$$

where $\alpha, \beta, \gamma$ and $\delta$ control the contribution of each terms to the overall loss. The required derivatives of $u$ and $\sigma$ w.r.t coordinates $x$ and $y$ are calculated via reverse mode differentiation provided by autodiff frameworks such as PyTorch or Tensorflow. Following Bar & Sochen (2021), we penalize violations in $\mathcal{L}_{\text{PDE}}^d$ twice using $L_2$ and $L_\infty$ norm variants in order to provide a stronger approximate solution. Finally, in order to train $u$-Net more efficiently, we enforce additional regularizers:

$$\mathcal{L}_{\theta_{u\text{-Net}}} = \mathcal{L}_\theta + \frac{\epsilon}{|\partial \Omega_e|} \sum_{e \in \{\partial \Omega_e\}} |u_e - f_e| + \zeta \|w_u\|^2 \tag{7}$$

The second term of Eq. 7 enforces Dirichlet BC only on electrodes while the last term controls the weights of $u$-Net using a weighed $L_2$ regularization on $\theta_u$ with temperature $\zeta$.

We train $\sigma$-Net (parameters are denoted as $\theta_\sigma$ such that it learns the conductivity distribution over all given mesh points to solve the semi-inverse problem. Similar to the forward problem training, $\sigma$-Net incorporates the main PDE and the Nuemann BCs seen in Eq. 6 as a part of its training objective in order learn the conductivity inside $\Omega$. However, the problem is known to be illposed and thus needs strong regularizers to improve the quality of reconstructions in conjunction with $\mathcal{L}_\theta$. For instance, we regulate the norm of gradients $\nabla_{x,y} \sigma$ inside $\Omega$ to promote sparse edges in predictions, and we penalize any conductivity prediction of less the one (= conductivity of vacuum) using $\mathcal{L}_{\text{hinge}}^h = \max(0, 1 - \sigma_h) \quad \forall h \in \{\Omega \cup \partial \Omega\}$.

The combined $\sigma$-Net training objective assimilating the elliptical PDE, its BC, and all the aforementioned mentioned regularizers is given as follows:

$$\mathcal{L}_{\theta_{\sigma\text{-Net}}} = \mathcal{L}_\theta + \frac{1}{|\partial \Omega|} \sum_{b \in \partial \Omega_b} |\sigma_b - \sigma_{\partial \Omega_b}^*| + \frac{\tau}{|\Omega|} \sum_{d \in \Omega} |\nabla_{x,y} \sigma_d| + \frac{\upsilon}{|\Omega \cup \partial \Omega|} \sum_{h \in \{\Omega \cup \partial \Omega\}} \mathcal{L}_{\text{hinge}}^h + \zeta \|w_\sigma\|^2, \tag{8}$$

where $\sigma_{\partial \Omega_b}^*$ is the known conductivity on the boundary points.

## 3 ENERGY BASED PRIORS

$\mathcal{L}_{\text{PDE}}^d$ from Eq. 6 can be interpreted as a residual of violation (of Eq. 1), noted as $r$ for simplicity. Assume $\mathbf{r}$ (residual of entire domain) follow a multi-variate Gaussian distribution with a zero mean and diagonal covariance matrix. Therefore, maximizing the likelihood of $p(\mathbf{r}|\boldsymbol{\sigma}; \theta_\sigma)$ results in the similar optimization as minimizing $\sum_{d \in \Omega} (\mathcal{L}_{\text{PDE}}^d)^2$ with respect to $\theta_\sigma$ (parameters of $\sigma$-Net). Now we can define our Bayesian approach but assuming $\boldsymbol{\sigma}$ follows the prior distribution $p(\boldsymbol{\sigma})$:

$$\max_{\theta_\sigma} \log p(\mathbf{r}|\boldsymbol{\sigma}) + \log p(\boldsymbol{\sigma}), \tag{9}$$

where $\boldsymbol{\sigma}$ is parameterized by $\theta_\sigma$ via $\sigma$-Net. We interpret $p(\boldsymbol{\sigma})$ as possible solutions to the PDE defined by Eq. 1 for different boundary conditions. We then define the prior distribution using energy-based models (EBMs) (LeCun et al., 2006): $(p(\boldsymbol{\sigma}) \propto \exp(-E_\phi(\boldsymbol{\sigma}))$. Several techniques

have been proposed for training EBMs, including contrastive divergence (Hinton, 2002), noise-contrastive estimation (Gutmann & Hyvärinen, 2010), score matching (Hyvärinen & Dayan, 2005), and denoising score matching (Vincent, 2011). We found that denoising score matching (DSM) is more stable, less compute intensive as it avoids expensive second order partial derivatives involved in score matching, and ultimately generates more realistic $\sigma$ solutions in our setting. DSM trains the energy function such that its vector field $(\nabla_{\boldsymbol{\sigma}_m} \log p_\phi(\boldsymbol{\sigma}_m))$ matches the vector field of the underlying distribution $p(\boldsymbol{\sigma}_m)$, which is approximated by perturbing the empirical data distribution with Gaussian noise of different intensities. See Song & Ermon (2019), Song & Ermon (2020) for more details.

We jointly estimate a noise conditional energy function $E_\phi(\boldsymbol{\sigma}, \mu_i)$ for all noise-perturbed data distributions conditioned on various noise scales $\mu_i$ for $i \in [1 \dots L]$, which are chosen such that $\mu_1 > \mu_2 > \cdots > \mu_L$. In our work, we chose $L = 20$ linearly spaced noise scales: $\mu_i \in [2, 0.01]$. Essentially, the training objective is to minimize the following:

$$\mathcal{L}(\phi; \mu_i) = \frac{1}{L} \sum_{i=1}^{L} \lambda(\mu_i) \left[ \frac{1}{2} \mathbb{E}_{p(\sigma)} \mathbb{E}_{\mathbf{z} \sim \mathcal{N}(0,I)} \left\| \nabla_\sigma E_\phi(\hat{\boldsymbol{\sigma}}, \mu_i) - \frac{\mathbf{z}}{\mu_i} \right\|_2^2 \right],$$ (10)

where $\lambda(\mu_i) > 0$ is a coefficient function chosen as $\lambda(\mu) = \mu^2$ and finally, $\hat{\boldsymbol{\sigma}} = \boldsymbol{\sigma} + \mu_i \mathbf{z}$ is the noise perturbed version of conductivity distribution $\hat{\boldsymbol{\sigma}} \sim \mathcal{N}(\boldsymbol{\sigma}, \mu_i^2 \mathbf{I})$. In order to use $E_\phi$ as a prior, in contrast to the standard DSM training that trains a score network $(S(.) = -\nabla_z E)$, we directly train the energy network. See Appendix A.3 for the details on training $E_\phi$ and also for generated samples from $E_\phi$ and their closest training data points (Fig. 6).

Upon successful training of the $E_\phi$ using Eq. 10, the energy function $E_\phi^*(\boldsymbol{\sigma}, \mu_L)$ will assign low energy values to $\boldsymbol{\sigma}$ that are more likely to present in the real world (valid solution of Eq. 1) and high energy value to unlikely assignments (invalid assignments that violates Eq. 1 greatly).

Now we can rewrite the final training objective of $\sigma$-Net (using Eq. 8) as follows:

$$\mathcal{L}_{\theta_{\sigma\text{-Net}}} = \mathcal{L}_\theta + \frac{1}{|\partial\Omega|} \sum_{b \in \partial\Omega_b} |\sigma_b - \sigma_{\partial\Omega_b}^*| + \frac{\tau}{|\Omega|} \sum_{d \in \Omega} |\nabla_{x,y} \sigma_d|$$
$$+ \frac{\upsilon}{|\Omega \cup \partial\Omega|} \sum_{h \in \Omega \cup \partial\Omega} \mathcal{L}_{\text{hinge}}^h + \zeta \|w_\sigma\|^2 - \kappa E_\phi^*(\boldsymbol{\sigma}, \mu_L)$$ (11)

where $\kappa$ is the weight of the prior in the overall loss.

## 4 EXPERIMENTS

Our EIT data simulation setup primarily consists of phantom generation and forward solution construction via finite element solver. Initially, we construct various discretized solutions of $\sigma$ on a 2D mesh-grid of size $128 \times 128$ to obtain phantoms $\Omega_{1\dots Z}$. In particular, we randomly choose the anomaly location, radius, shape deformity, quantity and their conductivity on a circular mesh. The target $\sigma$ values are chosen randomly between $\in [3, 15]$ for either 1, 2 or 3 anomalies per mesh with restrictions on their locations, so as to not let the anomalies touch each other and the boundary. We also assume a uniform background conductivity of $\sigma = 1$ (simulating vacuum) on locations devoid of any anomaly and $\sigma = 0$ outside the circle mesh. Upon selection of phantom configuration followed by discretization, we smooth our solutions following Evans (2010) using a Gaussian low-pass filter of size 200 and standard deviation 3 to obtain smoother solutions. We then generate 6512 such smoothed phantoms for EB prior training and 1628 for testing. Majority of these datasets contain circular anomalies with varying radii and a few thousand examples with minor shape deformity (ellipsoids). Furthermore, we standardize both train, test sets to $[0, 1]$ interval by dividing all samples using the maximum conductivity $\sigma$ value obtained from training set. We additionally generate a few hand-crafted phantoms to train the $u$-Net and $\sigma$-Net for forward and inverse problems, seen in the first row of Fig. 2. These configurations are specifically designed to be more challenging by setting higher shape deformity, larger variance in $\sigma$ values for each anomaly, and placing them in challenging locations inside the mesh. This is to purposefully make the physics-based training and inference more challenging to ensure extreme robustness. Note that there are no phantoms in the EBM training set which resemble these hand-crafted samples. With regards to our forward problem simulation setup, we assume that Neumann condition is imposed on the entire boundary where the function $g$ in Eq. 2 is given as a trigonometric pattern (Siltanen et al., 2000) as follows: $g = \frac{1}{\sqrt{2\pi}} \cos(\eta\omega + \psi)$, $n \in \mathbb{Z}$, where $\omega$ is the angle along $d\Omega$, and $\eta$ and $\psi$ are the current frequency and phase, respectively. We use one current pattern where $n = 1$ and $\psi = 0$ for all our experiments. Thus, to obtain the forward solution, we use finite element method (FEM) solvers by utilizing discretized hand-crafted

phantoms profile ($\sigma$ value, anomaly location) and current $g$ as input. We then solve for $u$ everywhere and obtain the solver based solution on circular mesh.

## 4.1 EXPEDITED SEMI-INVERSE PROBLEM EVALUATION

We now present results of our proposed framework. As discussed in prior sections, at each step of $\sigma$-Net training procedure, the current prediction $\hat{\sigma}$ is fed to the trained EB prior $E_\phi^*(\hat{\sigma}, \mu_L)$ to obtain a scalar energy value. This energy value provides useful supervision to $\sigma$-Net and essentially expedites the convergence of PINNs. This phenomenon can be viewed in Fig. 1. Here, each sub-figure represents the learning curve of $\sigma$-Net with the choice of metric as mean squared error (MSE), separately trained to recover conductivity of various phantoms with and without the inclusion of EB prior. Although the model is trained for 3000 epochs, for brevity, we only showcase the first 500 epochs of the curve due to more interesting properties here. Evidently, the PINNs with EB prior converge much faster within the first few 100 epochs than the training without prior. In case of phantom 4, the convergence is more than ten times faster while also aiding the PINN to avoid getting stuck in a local minima. Corresponding to these learning curves, we present Fig. 2 showing the quality of reconstruction for each phantom.

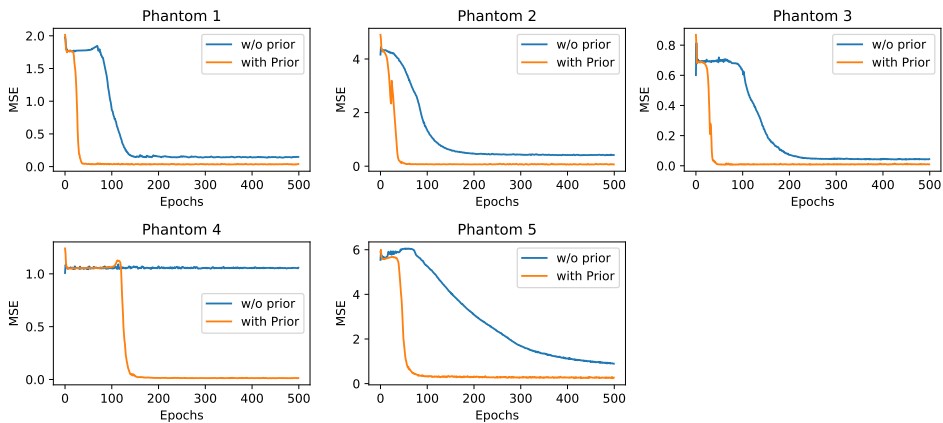

Figure 1: PDE solution convergence and accuracy analysis via $\sigma$ MSE for various phantoms.

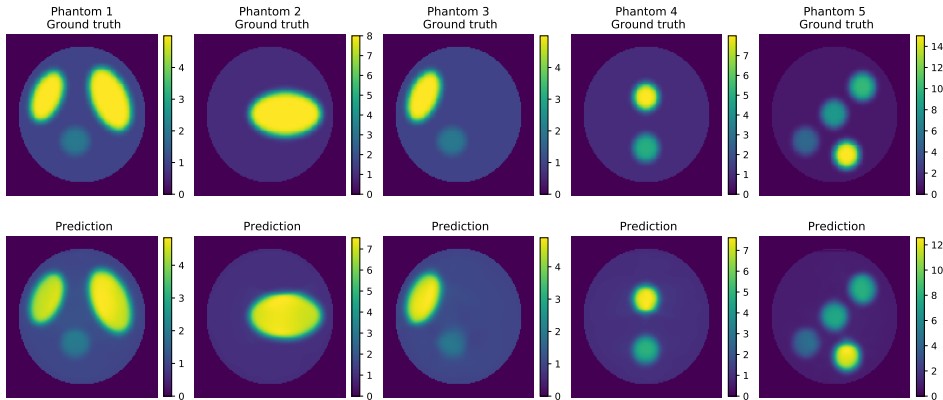

Figure 2: Comparison of ground truth $\sigma$ and predictions from PINN with EB prior.

## 4.2 COMPARISON WITH BASELINES

We now compare the performance of our framework with baselines Bar&Sochen Bar & Sochen (2021), Deep Galerkin Methods (DGM) Sirignano & Spiliopoulos (2018) by using the hand-crafted phantoms. In order to introduce these works, there are a few differences that set our works apart. Firstly, both Bar & Sochen (2021), Sirignano & Spiliopoulos (2018) utilized high resolution mesh which naturally enable PINNs to learn the problem more efficiently due to availability of large input data points to solve the PDE and BC. Additionally,

Bar & Sochen (2021) utilizes a large number of randomly selected boundary voltages to enforce the Dirichlet BC in Eq. 2 during forward solution learning. In contrast, we train our $u$-Net and $\sigma$-Net in a much more challenging settings while using a low-resolution mesh and only 16 boundary voltages during the forward problem training. The driving point here is that, one can view Dirichlet BC as a $L_1$ regression loss over given boundary voltages. When more boundary points and their corresponding voltages are revealed for learning $u$, the more stronger this supervision signal, which enables learning the forward map and anomaly location more accurately. However from practical EIT system standpoint, it is unrealistic to collect such close proximity fine-grained information over many boundary points. Apart from these difference, we treat Bar & Sochen (2021) and Sirignano & Spiliopoulos (2018) as special cases of our work. Our work primarily differs in the incorporation of the energy scalar from EBM and the new hinge loss term in Eq. 8 to improve the performance. Given the high non-linearity and ill-posed nature of the EIT problem, even a slight change in one parameter can lead to highly varying solutions. Hence, we tune our loss penalties differently to obtain higher quality reconstructions. We show a list of common parameters among the considered baselines. We now present the

Table 1: List of common hyperparameters among various baselines

| Method | Hyper-parameters | | | | | | | | | |
|---|---|---|---|---|---|---|---|---|---|---|
| | $\alpha$ | $\beta$ | $M$ | $\gamma$ | $\delta$ | $\epsilon$ | $\zeta$ | $\tau$ | $\nu$ | $\kappa$ |
| DGM | 1 | 0 | 0 | 1 | 1 | 1 | 0 | 0 | 0 | 0 |
| Bar&Sochen | 0.01 | 0.01 | 40 | 1 | 1 | 1 | 1e-8 | 0.01 | 0 | 0 |
| Ours | 0.05 | 0.05 | 40 | 1 | 0.1 | 100 | 1e-6 | 0.01 | 10 | 0.0001 |

Table 2: Evaluation of our method, DGM (Sirignano & Spiliopoulos, 2018), and Bar & Sochen (2021) with and without proposed EB Priors.

| Phantom | Metric | DGM | DGM w/ EB Prior | Bar&Sochen | Bar&Sochen w/ EB Prior | Our Method | Our Method w/ EB Prior |
|---|---|---|---|---|---|---|---|
| $\Omega_1$ | MSE $\downarrow$ | $3.26 \pm 0.028$ | $0.13 \pm 0.002$ | $2.57 \pm 0.041$ | $0.16 \pm 0.026$ | $0.13 \pm 0.001$ | $\mathbf{0.03 \pm 0.001}$ |
| | PSNR $\uparrow$ | $8.84 \pm 0.038$ | $22.77 \pm 0.079$ | $9.89 \pm 0.068$ | $22.02 \pm 0.645$ | $22.79 \pm 0.034$ | $\mathbf{29.94 \pm 0.146}$ |
| | MDE $\downarrow$ | $2.29 \pm 0.024$ | $0.36 \pm 0.003$ | $1.84 \pm 0.028$ | $0.23 \pm 0.002$ | $0.35 \pm 0.002$ | $\mathbf{0.12 \pm 0.014}$ |
| $\Omega_2$ | MSE $\downarrow$ | $6.19 \pm 0.021$ | $0.3 \pm 0.004$ | $5.18 \pm 0.039$ | $0.2 \pm 0.015$ | $0.4 \pm 0.002$ | $\mathbf{0.06 \pm 0.001}$ |
| | PSNR $\uparrow$ | $10.15 \pm 0.014$ | $23.3 \pm 0.054$ | $10.92 \pm 0.033$ | $25.00 \pm 0.325$ | $22.04 \pm 0.023$ | $\mathbf{30.58 \pm 0.079}$ |
| | MDE $\downarrow$ | $2.85 \pm 0.015$ | $0.39 \pm 0.003$ | $2.2 \pm 0.023$ | $0.25 \pm 0.002$ | $0.43 \pm 0.003$ | $\mathbf{0.1 \pm 0.009}$ |
| $\Omega_3$ | MSE $\downarrow$ | $1.56 \pm 0.032$ | $0.03 \pm 0.001$ | $1.37 \pm 0.017$ | $0.11 \pm 0.013$ | $0.04 \pm 0.001$ | $\mathbf{0.01 \pm 0.0001}$ |
| | PSNR $\uparrow$ | $12.05 \pm 0.089$ | $29.03 \pm 0.193$ | $12.62 \pm 0.056$ | $23.45 \pm 0.47$ | $28.2 \pm 0.061$ | $\mathbf{35.46 \pm 0.088}$ |
| | MDE $\downarrow$ | $1.38 \pm 0.033$ | $0.25 \pm 0.002$ | $1.09 \pm 0.021$ | $0.2 \pm 0.002$ | $0.25 \pm 0.002$ | $\mathbf{0.11 \pm 0.009}$ |
| $\Omega_4$ | MSE $\downarrow$ | $2.00 \pm 0.013$ | $0.08 \pm 0.002$ | $1.64 \pm 0.019$ | $0.2 \pm 0.036$ | $0.07 \pm 0.001$ | $\mathbf{0.01 \pm 0.0001}$ |
| | PSNR $\uparrow$ | $15.04 \pm 0.028$ | $29.07 \pm 0.099$ | $15.92 \pm 0.052$ | $25.23 \pm 0.779$ | $29.68 \pm 0.034$ | $\mathbf{38.45 \pm 0.052}$ |
| | MDE $\downarrow$ | $1.47 \pm 0.013$ | $0.26 \pm 0.004$ | $1.1 \pm 0.02$ | $0.19 \pm 0.001$ | $0.25 \pm 0.003$ | $\mathbf{0.12 \pm 0.011}$ |
| $\Omega_5$ | MSE $\downarrow$ | $7.39 \pm 0.046$ | $1.38 \pm 1.25$ | $6.68 \pm 0.018$ | $0.5 \pm 0.084$ | $0.57 \pm 0.003$ | $\mathbf{0.22 \pm 0.004}$ |
| | PSNR $\uparrow$ | $14.83 \pm 0.027$ | $24.03 \pm 1.99$ | $15.27 \pm 0.012$ | $26.67 \pm 0.694$ | $25.95 \pm 0.025$ | $\mathbf{30.17 \pm 0.082}$ |
| | MDE $\downarrow$ | $2.97 \pm 0.012$ | $0.73 \pm 0.439$ | $2.47 \pm 0.017$ | $0.28 \pm 0.004$ | $0.44 \pm 0.002$ | $\mathbf{0.08 \pm 0.016}$ |

semi-inverse problem evaluation results for considered baselines in Table 2 for all five phantoms. We primarily use three metrics namely, mean squared error (MSE), peak signal-to-noise ratio (PSNR) and mean difference error (MDE) to evaluate the quality of reconstruction of $\sigma$ solution while learning the objective Eq. 8. Note that MDE is computed as the difference in means of $\sigma$ over all $(x, y) \in \Omega$ and $d\Omega$. We then utilize the hyper-parameters shown in Table 1 to train $\sigma$-Net for our framework, while the neural network presented in Bar & Sochen (2021) is used on both Bar & Sochen (2021) and Sirignano & Spiliopoulos (2018) baselines. All these models are trained separately on all five phantoms while using the same settings, once without the EB prior and once under the presence of EB prior. These experiments are repeated for 10 runs and the averaged metrics along with their standard deviation error is presented. The main observation from Table 2 is, that by incorporating the EB prior in PINN training enables lower error rates and higher quality reconstructions for the semi-inverse problem. Additionally, we can also see that the EB prior can be easily plugged into any PINN-based framework and vastly improve the underlying training, showcasing our methods strong generalizability.

## 4.3 LEARNING RATE CONVERGENCE

We aim to show through this study that the learning rate vastly affects PINN's solution accuracy and this inherent instability can be easily alleviated by introducing EB priors, even while using large learning rates. For this study, we keep all the parameters of our proposed method shown in Table 1 fixed and vary only the learning rate during $\sigma$-Net training with Eq. 8. We choose 0.0001, 0.001, 0.01 and 0.1 learning rates to study their effects on the semi-inverse PINN training with and without the EB prior. Note that, we still decay all these chosen learning rates using exponential decay with rate 0.9 over every 200 epochs. The results of learning rate studies are presented in Fig. 3. As seen in this figure, the true $\sigma$ value for the ellipsoid anomalies is 5 while the

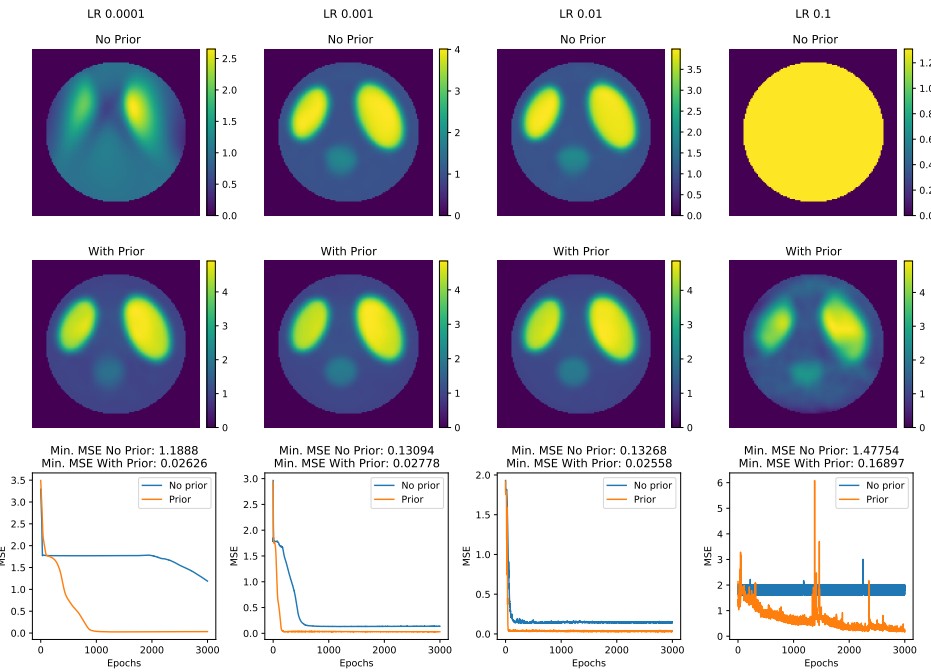

Figure 3: Training robustness of our proposed work with EBM prior for various learning rates on Phantom 1.

smaller circular anomaly in the bottom-middle has $\sigma = 2$. We can evidently see that the PINN without EB prior fails to converge to an accurate solution with regards to row three. More importantly, the maximum $\sigma$ values for row one and two's color-bars indicate the failure to reach an accurate conductivity prediction. Clearly, the PINN without EB prior never matches the solution accurately, under all learning rates. More adversely, the PINN produces a trivial solutions while using a learning rate that is either too low or too high. On the other hand, PINNs augmented with EB prior always produce accurate solutions of slightly varying degree of MSE as seen in row two, three of Fig. 3. More importantly, the training session with extremely large learning rate $LR = 0.1$ still manages to produce a reconstruction although the accuracy might be lower. This proves the robustness of our framework and instigates clear evidence that PINN training is being improved by a very large margin while incorporating EB priors. Other advantages such as, burden of choosing the optimal learning rate is also alleviated.

## 4.4 SENSITIVITY ANALYSIS VIA RANDOM PARAMETER SEARCH

As discussed in these earlier sections, PINNs are extremely sensitive to the balance of interplay among different hyperparameters. The training procedure essentially becomes an optimization problem where one has to tune the right settings in order to converge to a highly accurate solution or obtain a failed trivial solution. The problem of choosing parameters is even more challenging in EIT as it is highly non-linear and ill-posed. We thus study the effect of changing the strength of various loss weighting penalties mentioned in Table 1 while simultaneously observing the sensitivity of the EB prior trained with various data quantities. As a preliminary, we first bifurcate the 6512 training samples into a smaller training, validation sets (20% set aside for cross-validation). After splitting, we separately train five EB priors (labeled as 100% data priors) on the whole training set. In addition, we train another five EB priors (labeled as 1% data priors) separately, by randomly selecting only 1% of the data that was used for training the 100% data priors. Note that the model architecture and all hyper-parameters are fixed for consistency, while only the batch sizes are different; 64 and 16 for 100% and 1% data priors respectively (see more training details and synthetic generated samples in Appendix A.4).

The final step is to evaluate all the trained EBMs and perform sensitivity analysis of PINN training. For this, we restrict the search space to $\alpha$, $\beta$, $\gamma$ and $\delta$ parameters in Eq. 6 as they are the main terms influencing the solution quality of the EIT inverse problem. We randomly choose a single number from each of the search configuration set and start training the PINN: $\alpha \in [0.01, 0.05, 0.1, 0.5, 1]$, $\beta \in [0.01, 0.05, 0.1, 0.5, 1]$, $\gamma \in [0.01, 0.1, 0.5, 1.0, 2.0]$ and finally $\delta \in [0.001, 0.01, 0.1, 0.5, 1.0, 2.0]$. We use Phantoms 1 and 5 to conduct 100 random search runs for PINN with each of the ten EBMs as priors, and 500 runs for PINN without EB prior. We then average the results of every training category (w/o prior, 1% and 100% prior) and display them

in Fig. 4. Here, the y-axis in Fig. 4a indicates that a large number of random runs were successful to solve Phantom 1 and reach a set upper bound of 0.3 MSE and 3000 epochs. Fig. 4b also corresponds to Phantom 1, however we set a strict MSE threshold of 0.3 and check for number of experiments that reach this threshold by the end of all 3000 epochs. Similar observations can be said about Figs. 4c and 4d which are studies related to Phantom 5. In conclusion, we have substantial evidence that the PINN with EB prior always has higher success rates in producing highly accurate $\sigma$ reconstructions. More importantly, the PINN with the EB prior converges much faster towards near-zero MSEs regardless of quantity of the EBM training sample sizes. Thus, showcasing the strong robustness of our proposed framework.

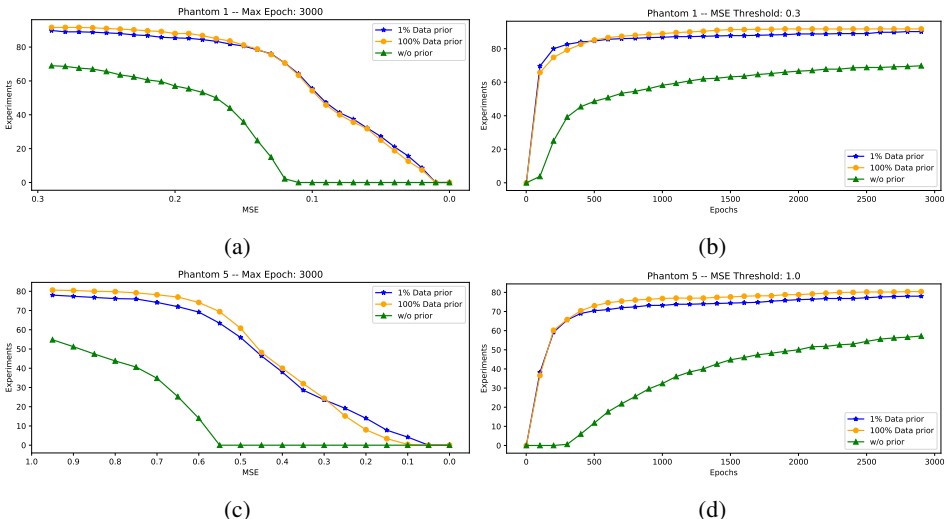

Figure 4: Sensitivity analysis of $\sigma$-Net while randomly searching the optimal weights in Eq. 6.

## 4.5 SEMI-INVERSE WITH NOISY DATA

In the final set of experiments, we test the robustness of our framework while learning from noisy data. Initially, we add uniform noise of various standard deviation levels such as $0.1, 0.25, 0.5$, to corrupt the 16 boundary measurements. The $u$-Net learns the forward problem under these noisy settings and approximates a noisy-forward map (see Appendix A.2 for evaluation of $u$-Net under noisy settings). While moving to the semi-inverse problem, we use data originating from the noisy $u$-Net predictions and check $\sigma$-Net's ability to withstand measurement noise. These results can be seen in Table 3. The performance degradation is visible when noise std increases, more noticeable in some phantoms that others. Despite these noisy conditions, incorporating the EB prior during $\sigma$-Net training greatly improves the predictions and robustness.

Table 3: Performance evaluation of $\sigma$-Net using noisy measurements

| Noise | Metric | Phantom 1 | | Phantom 2 | | Phantom 3 | | Phantom 4 | | Phantom 5 | |
|---|---|---|---|---|---|---|---|---|---|---|---|
| | | w/o Prior | w/ Prior | w/o Prior | w/ Prior | w/o Prior | w/ Prior | w/o Prior | w/ Prior | w/o Prior | w/ Prior |
| 0 | MSE ↓ | 0.13 | **0.03** | 0.4 | **0.06** | 0.04 | **0.01** | 0.07 | **0.01** | 0.57 | **0.22** |
| | PSNR ↑ | 22.79 | **29.94** | 22.04 | **30.58** | 28.20 | **35.46** | 29.68 | **38.45** | 25.95 | **30.17** |
| | MDE ↓ | 0.35 | **0.12** | 0.43 | **0.1** | 0.25 | **0.11** | 0.25 | **0.12** | 0.44 | **0.08** |
| 0.1 | MSE ↓ | 0.094 | **0.03** | 3.38 | **1.64** | **0.02** | 0.03 | 2.84 | **0.01** | 0.49 | **0.11** |
| | PSNR ↑ | 24.25 | **29.91** | 12.77 | **15.91** | 31.50 | 29.72 | 13.53 | **36.78** | 26.67 | **33.09** |
| | MDE ↓ | 0.29 | **0.16** | 1.09 | **0.52** | 0.21 | **0.21** | 0.02 | **0.17** | 0.42 | **0.23** |
| 0.25 | MSE ↓ | 0.17 | **0.02** | 0.58 | **0.08** | 0.03 | **0.03** | 0.09 | **0.03** | 0.67 | **0.31** |
| | PSNR ↑ | 21.55 | **31.20** | 20.40 | **29.10** | 28.95 | **29.29** | 28.46 | **33.86** | 25.24 | **28.56** |
| | MDE ↓ | 0.37 | **0.23** | 0.50 | **0.29** | 0.24 | **0.16** | 0.26 | **0.17** | 0.49 | **0.37** |
| 0.5 | MSE ↓ | 0.13 | **0.06** | 0.37 | **0.13** | 0.61 | **0.57** | 0.09 | **0.02** | 5.22 | **4.39** |
| | PSNR ↑ | 22.96 | **26.05** | 22.42 | **26.84** | 16.15 | **16.41** | 28.69 | **35.43** | 16.35 | **17.09** |
| | MDE ↓ | 0.34 | **0.13** | 0.42 | **0.13** | 0.14 | **0.22** | 0.25 | **0.17** | 0.90 | **0.34** |

## 5 RELATED WORK

Inverse problems for elliptic PDEs are ill-posed and highly nonlinear, which makes it imperative to include regularization methods for obtaining a reasonable reconstruction Jin et al. (2012); Jin & Maass (2012). Many

approaches exit for solving the EIT inverse problem with rigorous theoretical justifications. For e.g., variational methods for least-square fitting (Jin et al. (2012); Jin & Maass (2012); Ahmad et al. (2019)), the factorization method (Kirsch & Grinberg (2008)), the d-bar method (Mueller et al., 2002; Isaacson et al., 2004; Knudsen et al., 2009), the monotonicity shape estimate Harrach & Ullrich (2013), the level set method Liu et al. (2017; 2018) and series inversion methods Arridge et al. (2012); Abhishek et al. (2020) to name a few. Besides, regularized Newton type methods such as Haber (2005); Lechleiter & Rieder (2008) are also popular. For the groundwater flow model, some important references can be found in Frühauf et al. (2005); Russell & Wheeler; Oliver et al. (2008).

Driven by the need to quantify uncertainties, many works proposed to solve such inverse problems in a parametric Bayesian setting (Kaipio et al. (2000; 2004); Strauss & Khan (2015); Ahmad et al. (2019); Strauss et al. (2015)). However, the error in these methods rely on discretization schemes hence, alternative non-parametric Bayesian inversion mesh-independent methods have been proposed. Following Stuart (2010), a Bayesian level set method was proposed for reconstruction of piecewise smooth conductivity in Dunlop & Stuart (2016), for reconstruction of piecewise continuous permeability in ground water flow model in Iglesias et al. (2014) and for reconstruction of conductivity in certain Sobolev spaces in Abraham & Nickl (2019). Recall that in a Bayesian setting, the posterior $p(\theta|\sigma) \propto p(\theta)p(\sigma|\theta)$ depends on the space of conductivities $\sigma$ and a prior. The prior $p(\theta)$, in addition to encapsulating our subjective belief about the functional space; also introduces regularization in the problem. Great strides have been made in using efficient MCMC sampling methods which increased computational efficiency of such Bayesian inversion methods (Cotter et al., 2013).

With the rise of machine learning methods, many solutions for solving the EIT with neural networks were proposed. In a supervised setting, Hamilton & Hauptmann (2018); Fan & Ying (2020); Hamilton et al. (2019); Agnelli et al. (2020) primarily use convolutional neural network (CNN) architectures to predict the conductivity $\sigma$ based on boundary measurements. Deviating to other inverse problems, Ulyanov et al. (2018); Lucas et al. (2018) describe the use of various generative models in this context. Generative Adversarial Networks (GANs) (Goodfellow et al., 2020) and variants are a great example. Patel & Oberai (2020); Patel et al. (2020; 2022); Molin (2022) solve various image and physics-inspired inverse problems by training sampling algorithms that jointly learn measurements $y$ and latent vectors $z$ with generator ($G$) as a prior. This allows for generation of new samples given the posterior, G and unseen $y$. In contrast, Marinescu et al. (2020) use pretrained generators as priors for maximum a-posteriori (MAP) inference over a given $z$, known corruption model $f$ and recover a corrupted input image $x$ such that $G(z^*) = \arg\max_z p(z)p(x|f \circ G(z))$, where $G(z^*)$ is the clean image. Lastly, Adler & Öktem (2018; 2019) trained conditional-WGANs in supervised setting (pair-wise data and $y$) and sample a posterior given conditional $G, z$, and unseen $y$. Their second approach allows for trainable point-wise inference networks similar to our work, but these rely on supervised data. In comparison, our work learns both EBMs and PINNs in an unsupervised manner directly in data space, resulting in a more direct and faster inference.

Instead of using implicit GAN priors via latent spaces, score and energy models are more direct due to stronger a-priori beliefs. In this work, we share a connection with score matching (SM) Hyvärinen & Dayan (2005) through DSM. Many score-based works solved *linear* inverse problems utilizing DSM (Ramzi et al. (2020); Song et al. (2021); Kawar et al. (2021); Chung & Ye (2022)). For MRI/CT reconstruction, Ramzi et al. (2020) score-prior with posterior sampling using annealed HMC; Song et al. (2021); Chung & Ye (2022) score-prior with posterior sampling using reverse-time stochastic differential equations; Kawar et al. (2021) score-prior with posterior sampling using LD and singular value decomposition (SVD); Kadkhodaie & Simoncelli (2021) generalize DSM and plug-and-play priors (Venkatakrishnan et al., 2013) and sample with an implicit CNN denoising prior. While, Zach et al. (2022) use EB priors trained with contrastive divergence and solve linear inverse problems by sampling with proximal gradient descent. Clearly, the similarities between our work and aforementioned are purely prior design choices; (1) DSM is used for training the prior; (2) all our works use discretized inverse map solutions (MRI/CT images or EIT $\sigma$-phantoms in our case) as input data; (3) there is no dependency on measurement data (labels) during prior training. The important difference is that we solve non-linear, ill-posed physics-based inverse problem of EIT using DSM priors without sampling the posterior.

## 6 CONCLUSION

Physics-informed neural networks are an important category of data-driven PDE solvers. However, PINNs are not stable and robust for more complicated problems. In this work, we look at the EIT problem and show that we can improve the stability and robustness of PINNs training (w.r.t noisy measurement and hyper-parameter selection) via a Bayesian approach. We describe a data-driven prior using energy-based methods, which can easily be used with the other loss terms and robustly train PINNs. In the EIT setting, our experimental results also show that PINNs converge faster and also to a more accurate solution when trained with prior. In our future work, we will study using priors for potential in order to the improve the training of PINN for full EIT inverse problem and other PDE problems.

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

## A  APPENDIX

The model architectures and training implementation details for $u$-Net, $\sigma$-Net and EBM are shared here.

### A.1  SEMI-INVERSE IMPLEMENTATION DETAILS

The common loss weighting penalties used by both the $u$-Net, $\sigma$-Net models are: $\alpha = 0.05$, $\beta = 0.05$, $M = 40$, $\delta = 0.1$ and $\zeta = 1e^{-6}$. The Nuemann loss penalty $\gamma = 0.1$ and Dirichlet loss penalty $\epsilon = 100$ are specific to the forward problem only. While, $\gamma = 1$, $\tau = 0.01$, $\nu = 10$ and lastly $\kappa = 0.0001$ are specific inverse problem. Additionally, both models use the same multi-layer perceptron (MLP) architecture with residual connections (Wang et al., 2021) and consist of 4 hidden layers with $\tanh$ activation and 64 neurons each. A single output neuron with no activation aids in predicting the solutions for given mesh points. We train both these models with a batch size of 1000 using ADAM (Kingma & Ba, 2014) optimizer for 3000 epochs with an initial learning rate of 0.005 and decay it exponentially with a rate of 0.9 over intervals of every 200 epochs. We primarily use NVIDIA Titan RTX GPU for all our training purposes and the training time for forward and semi-inverse problems separately take around takes $\sim$ 8 minutes for 3000 epochs.

### A.2  FORWARD PROBLEM IMPLEMENTATION AND RESULTS

Table 4 shows the evaluation results of $u$-Net for the forward problem on the considered phantoms. We study the quality of forward problem solutions from $u$-Net while adding random uniform noise to the 16 electrode measurements of various levels. As seen in these experiments, the quality degradation is obvious due to increasing levels of noise. To counter this effect, we tune the penalty weights $\alpha, \beta, \gamma, \delta$ on individual loss terms in Eq. 7 for increasing noise levels to improve the quality of reconstructions. The weights on Dirichlet BC $\epsilon = 100$, $L_2$ model parameter regularization $\zeta = 1e - 6$ are fixed for all experiments and phantoms. Additionally, we use the same parameter set for all phantoms while only tuning them for a given noise level, which showcases the robustness of our method.

Table 4: Performance evaluation of $u$-Net for solving Forward problem with various noise levels

| Noise Std | Metric | Phantom 1 | Phantom 2 | Phantom 3 | Phantom 4 | Phantom 5 |
|---|---|---|---|---|---|---|
| | | Parameters: $\alpha = 0.05$, $\beta = 0.05$, $\gamma = 0.1$, $\delta = 0.1$ | | | | |
| 0 | MSE $\downarrow$ | 0.00046 | 0.00091 | 0.00027 | 0.00058 | 0.00036 |
| | PSNR $\uparrow$ | 39.41 | 33.40 | 46.90 | 47.02 | 40.51 |
| | | Parameters: $\alpha = 0.1$, $\beta = 0.1$, $\gamma = 0.1$, $\delta = 0.1$ | | | | |
| 0.1 | MSE $\downarrow$ | 0.0090 | 0.070 | 0.02076 | 0.0208 | 0.026 |
| | PSNR $\uparrow$ | 26.51 | 14.55 | 28.13 | 31.48 | 21.91 |
| | | Parameters: $\alpha = 0.05$, $\beta = 0.05$, $\gamma = 1$, $\delta = 0.1$ | | | | |
| 0.25 | MSE $\downarrow$ | 0.0426 | 0.1665 | 2.667 | 1.722 | 0.039 |
| | PSNR $\uparrow$ | 16.70 | 13.854 | 7.04 | 12.30 | 20.22 |
| | | Parameters: $\alpha = 0.1$, $\beta = 0.05$, $\gamma = 1$, $\delta = 0.1$ | | | | |
| 0.5 | MSE $\downarrow$ | 1.041 | 2.25 | 10.59 | 2.46 | 4.70 |
| | PSNR $\uparrow$ | 5.89 | 0.52 | 1.05 | 3.01 | 0.587 |

### A.3  EB PRIOR IMPLEMENTATION AND EVALUATION

**Implementation**  We train our EB prior on $\sigma$ solutions (as described in Section 4) by perturbing them with Gaussian noise of 20 noise scales $\mu_i \in [2, 0.01]$ in order to learn by DSM training. We visualized some of the training samples in Fig. 7 and the compared the nearest samples in EBM training data and inference set of PINN in Fig. 5. We then create a deep convolutional neural network with multiple residual He et al. (2015) connections inspired by the architectures in Song & Ermon (2019) and Song & Ermon (2020). In particular, our model consists of a single $3 \times 3$ convolution layer followed by series of 16 convolutional residual and residual-downsampling blocks in the form of an encoder block of typical convolution encoders. Each of these blocks consist of convolutional and Group normalization Wu & He (2018) layers with a group size 32. We then learn the input features by gradually down-sampling them to smaller resolutions. Upon feature map reduction at the end of the last convolution layer, we introduce a single hidden dense layer of 256 units to jointly learn the compressed $\sigma$ features along with the noise scales, followed by a final dense layer which outputs scalar energy values. We use ELU non-linearity for all our EBM layers except the last energy layer which has no activation. We then train this EBM model to denoise the noise perturbed $\sigma$ solutions via the objective seen in Eq. 10 for 2000 epochs, with a batch size of 64, using ADAM optimizer with a fixed learning rate of 0.0001. The training time takes around 15 hours while using a NVIDIA Titan RTX GPU.

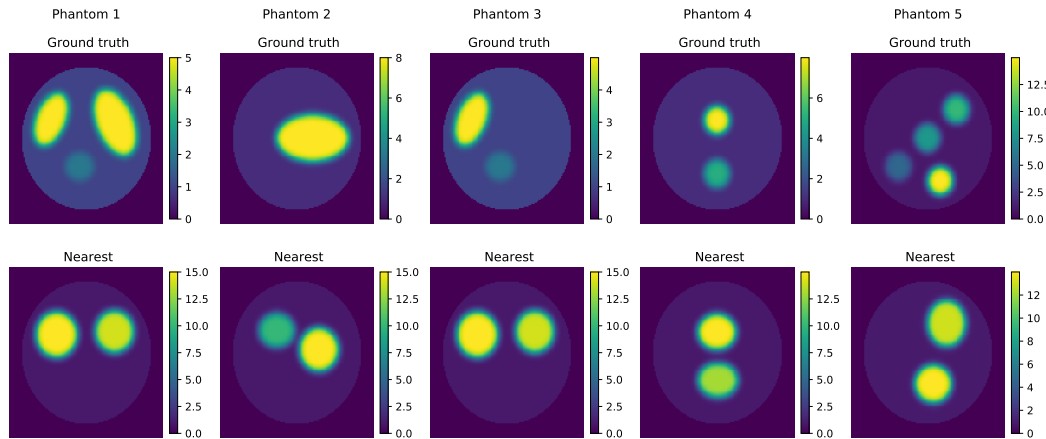

Figure 5: Comparison of ground truth $\sigma$ and nearest data in EBM training set

**Evaluation** After the EBM $E_\phi^*(\sigma, \mu)$ is trained, we can generate samples by using annealed Langevin dynamics (LD) Song & Ermon (2019) sampling. We start from a fixed prior distribution such as uniform noise and initially run LD for 100 steps with a step size of $s_{n_1}$, using first $\mu_1$ noise scale and draw samples from $E_\phi(., \mu_1)$ by adding Gaussian noise. Next we draw samples from $E_\phi(., \mu_2)$ by reducing the step size and refine the samples. We continue running LD until all the noise scales are used to sample and we finally arrive at the final step size $s_{n_L} = a\mu_i^2/\mu_L^2$, where $a = 0.0002$ in our inference run. During the final step of last noise scale $\mu_L$, we perform 1 step of gradient descent instead of LD to obtain an better denoised version of the $\sigma$ generations. In Fig. 6, we display some of these curated $\sigma$ solutions along with their nearest neighbors in the training set which are obtained by calculating the $L_2$ distance between all training set examples.

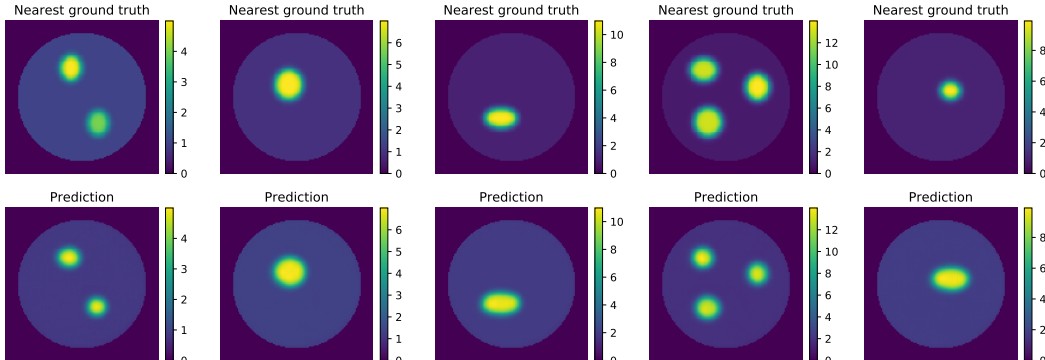

Figure 6: Comparison of training set samples to nearest generated $\sigma$ using annealed Langevin dynamics.

### A.4 ENERGY-BASED PRIOR SENSITIVITY ANALYSIS

In this section, we provide some additional details to understand EB prior sensitivity to the quantity of training data. For fairness in comparison w.r.t original experiments, we fix the EBM architecture and all training hyperparameters which were originally used for the main results mentioned in Appendix A.3, except for the batch size so as to cater for small training sample sizes and better generalization. We additionally employ early stopping with a patience of 100 epochs by monitoring the validation loss in order to avoid problems such as over-fitting and memorization of data. After training with DSM on all the EBMs, we sample from the best and worst energy models trained on the $1\%$ subset data using annealed Langevin dynamics and generate synthetic data shown in Figs. 8 and 9 respectively.

The ground truth training set samples are provided in Fig. 7 for visual comparison. As seen in these figures, despite being trained on such small sample sizes, the EBMs are capable of understanding salient features such as anomaly shape, anomaly conductivity $\sigma$ values. Moreover, these models even display the ability to infer

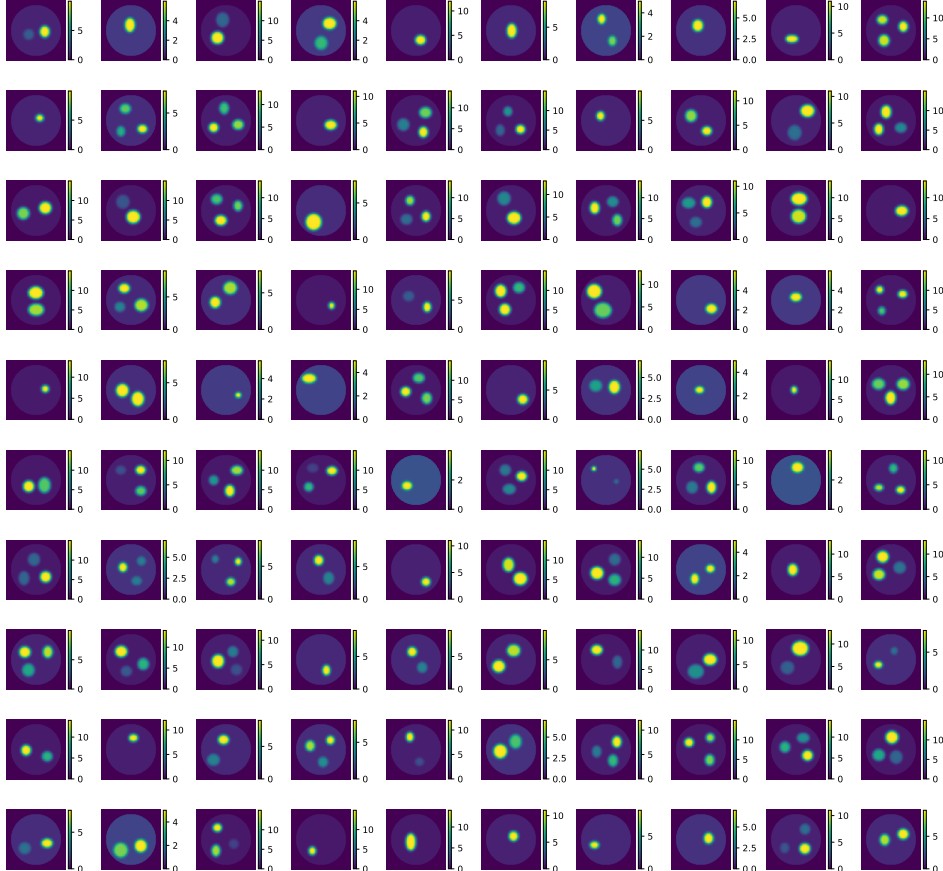

Figure 7: Ground truth data from training set

essential rules for our mathematical model such as, anomalies should not touch each or the circular boundaries. In case of the best $1\%$ data EBM in Fig. 8, it even shows capabilities to generate unseen modes which contains more than one, two or three anomalies modes in training set and of different shapes and conductivities.

## A.5 FULL-INVERSE EIT

We share some implementation details and evaluations of a much harder and highly ill-posed inverse problem of EIT. By problem definition, given a set of boundary voltages at $N_e$ electrode locations and its corresponding current $g$, the goal is to recover the target $\sigma$ and $u$ inside $\Omega$ jointly while all the networks are initiated from scratch. We train all PINN models jointly.

$$\mathcal{L}_U = \frac{1}{K} \sum_{k=1}^{K} \mathcal{L}_{\theta_{u_k\text{-Net}}} \tag{12}$$

$$\begin{aligned}
\mathcal{L}_{\theta_{\sigma\text{-Net}}} = \mathcal{L}_\theta &+ \frac{1}{|\partial\Omega|} \sum_{b\in\partial\Omega_b} |\sigma_b - \sigma^*_{\partial\Omega_b}| + \frac{\tau}{|\Omega|} \sum_{d\in\Omega} |\nabla_{x,y}\sigma_d| \\
&+ \frac{\upsilon}{|\Omega\cup\partial\Omega|} \sum_{h\in\Omega\cup\partial\Omega} \mathcal{L}^h_{\text{hinge}} + \zeta\,\|w_\sigma\|^2 - \kappa E^*_\phi(\boldsymbol{\sigma}, \mu_L),
\end{aligned} \tag{13}$$

As seen in Eq. 12 and Eq. 13, we use $K = 8$ which correspond to multiple current patterns $g$ from Eq. 2 in our joint training scheme. Here, each current pattern is learnt a separate $u$-Net like model and one $\sigma$-Net model learns the conductivity distribution inside $\Omega$. Essentially, we obtain $K$ loss values from $K$ current patterns which are then averaged and used to update every $u$-Net and $\sigma$-Net model simultaneously. We present some of the phantoms along with their $\sigma$-Net reconstructions used in evaluation of this joint training strategy in Fig.

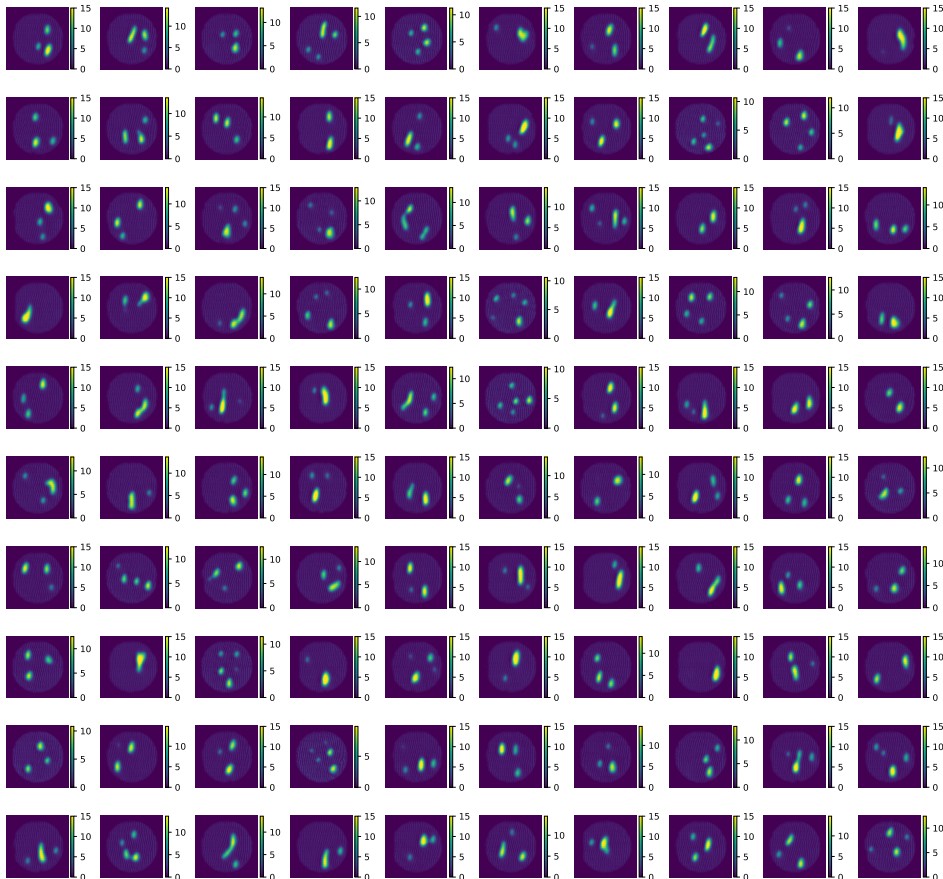

Figure 8: Uncurated synthetic samples generated from the best EBM trained on $1\%$ of the training set.

10. As seen in this figure, we are able to recover $\sigma$ accurately for simple anomalies like row one. However, when the phantom configuration becomes complex, the joint training fails. We reserve this for our future work in scope of improving this method further.

## A.6 FINITE ELEMENT METHOD BASELINE

We provide an additional baseline which is based on Finite Element Method (FEM) reconstructions. More particularly, we utilize Levenberg-Marquard (LM) (Horesh et al., 2007) which is an iterative second order gradient-based optimization algorithm and compare its performance against both our proposed semi-inverse and full-inverse EIT setups. The problem formulation according to LM is shown below:

$$\mathcal{L}_\sigma = \arg\min_\sigma \quad ||\mathcal{F}(\sigma) - BV_\sigma||^2 + \nu||w|| \tag{14}$$

Here in Eq. 14, $\mathcal{F}(\sigma)$ is a forward operator that maps the conductivity $\sigma$ distribution into the data space, $BV_\sigma$ are the given boundary measurements at designated electrode locations. And lastly, $w$ is a regularization operator with temperature $\nu$ which allows for inclusion of any prior information and generally encourages goodness of data fitting properties through the data residual. We then use an iterative approach to compute the gradient $g$ and Hessian $H$ to optimize the final objective function $\nabla$ as follows:

$$\nabla\gamma = H \ g \qquad \text{where;}$$
$$H = \mathbf{J}^T\mathbf{J} + \lambda_t\boldsymbol{I} + \nu\nabla^2 w \tag{15}$$
$$g = \mathbf{J}^T||\mathcal{F}(\sigma) - BV_\sigma||^2 - \nu\nabla w$$

The term $\mathbf{J}$ in Eq. 15 refers to the Jacobian matrix which is computed by $\partial BV_\sigma/\partial\sigma$, $\boldsymbol{I}$ is the identity matrix and $\lambda_t$ is a hyper-parameter to control the strength of Hessian. The LM method explicitly employs second

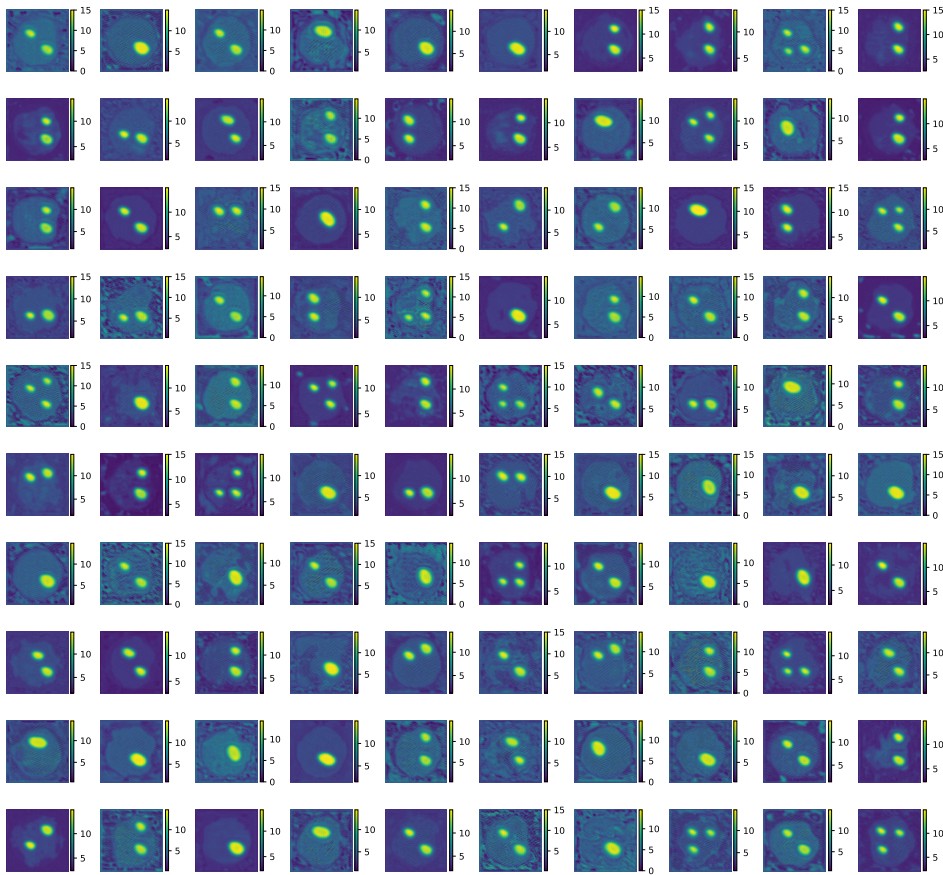

Figure 9: Uncurated synthetic samples generated from the worst EBM trained on 1% of the training set.

order derivatives inside the objective while using only the measurements on the electrodes to reconstruct the images. Hence, it is fair when LM is compared against our full-inverse EIT PINN as both methods use similar data to compute the final solution. We provide Fig. 10 and Table 5 in order to showcase the performance of our proposed approach. As seen here, our method outperforms LM on all metrics and achieves higher accuracy while pinpointing the location of anomaly. Lastly, although it is unfair to compare LM to semi-EIT as the semi-EIT problem has access to the data from the internal domain, we provide results on the phantoms displayed in Fig. 2 for the sake of completeness. The corresponding results are shown in Table 6 and Fig. 12.

Table 5: Comparison of Full-EIT augmented with EB Prior and Levenberg-Marquard FEM reconstructions from phantoms seen in Fig. 10.

| Phantom | Metric | Levenberg-Marquard | Full-EIT with EB Prior |
|---|---|---|---|
| $\Omega_1$ | MSE ↓ | 1.63 | **1.30** |
| | PSNR ↑ | 16.04 | **16.90** |
| | MDE ↓ | 0.10 | **0.022** |
| $\Omega_2$ | MSE ↓ | 0.36 | **0.11** |
| | PSNR ↑ | 18.46 | **23.53** |
| | MDE ↓ | 0.31 | **0.20** |
| $\Omega_3$ | MSE ↓ | 1.01 | **0.12** |
| | PSNR ↑ | 14.02 | **23.26** |
| | MDE ↓ | 0.26 | **0.22** |

## A.7 COMPUTATION COST AND MEMORY USAGE ANALYSIS

We now present the last set of results by comparing the computation cost of PINN training with and without inclusion of the EB prior. The histograms in Figs. 13 show the computation time (in seconds) using various

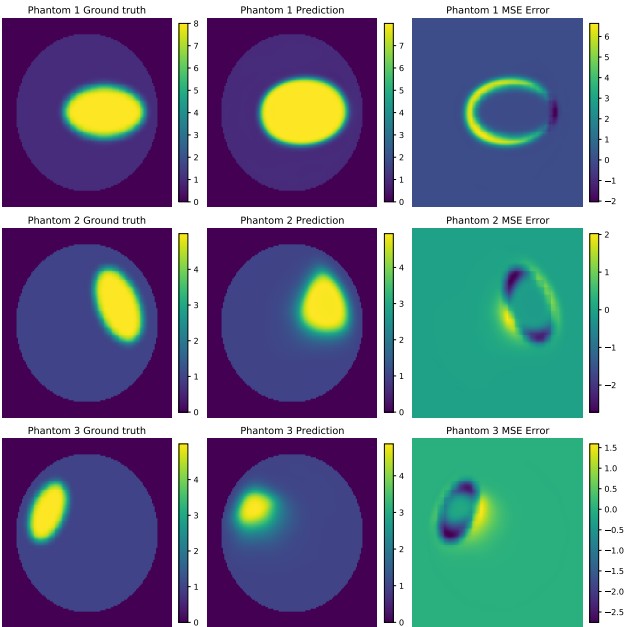

Figure 10: Unsupervised Full Inverse EIT evaluation on some considered phantoms.

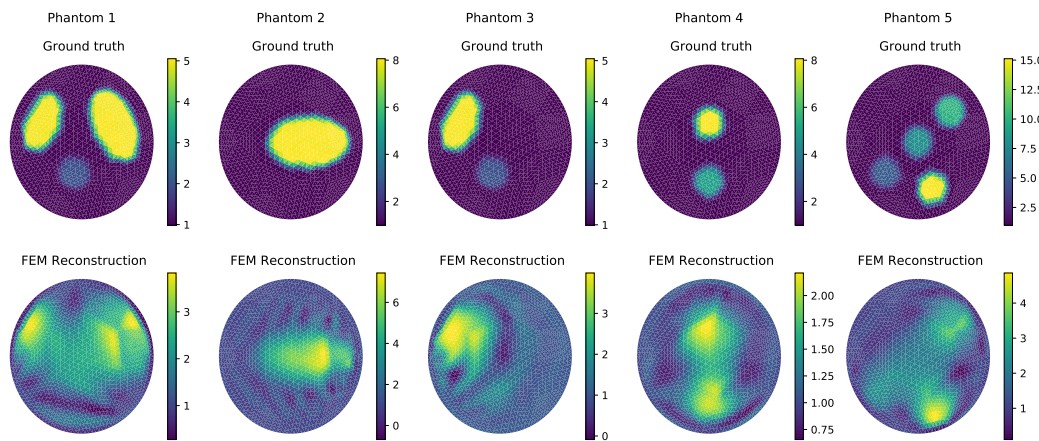

Figure 11: Comparison of ground truth phantoms used for Full-EIT in Fig. 10 and predictions from FEM-based iterative gradient Levenberg-Marquardt.

Nvidia GPUs along with the peak memory consumption (in MB) for 500 runs of PINN without prior and 500 runs of PINN with EB prior. It is straightforward to infer that the PINN without an EB prior is faster in computation time and uses less memory however, the numbers for our proposed framework are quite comparable. One on an average, the compute for PINN with EB priors is slightly less than twice that of normal PINN. Given the massive performance and stability guarantees, the meagre increase in time and compute cost is admissible.

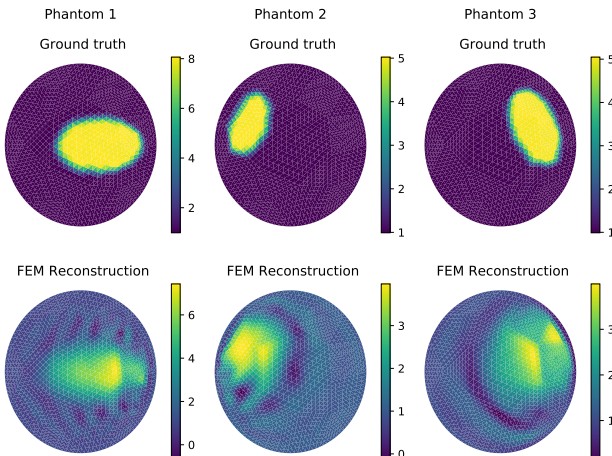

Figure 12: Comparison of ground truth phantoms used for Semi-EIT in Fig. 2 and predictions from FEM-based iterative gradient Levenberg-Marquardt.

Table 6: Comparison of Semi-EIT augmented with EB Prior and Levenberg-Marquard FEM reconstructions from phantoms seen in Fig. 2.

| Phantom | Metric | Levenberg-Marquard | Semi EIT with EB Prior |
|---------|--------|--------------------|------------------------|
| $\Omega_1$ | MSE $\downarrow$ | 1.59 | $0.03 \pm 0.001$ |
| | PSNR $\uparrow$ | 12.07 | $29.94 \pm 0.146$ |
| | MDE $\downarrow$ | 0.07 | $0.12 \pm 0.014$ |
| $\Omega_2$ | MSE $\downarrow$ | 1.63 | $0.06 \pm 0.001$ |
| | PSNR $\uparrow$ | 16.04 | $30.58 \pm 0.079$ |
| | MDE $\downarrow$ | 0.10 | $0.1 \pm 0.009$ |
| $\Omega_3$ | MSE $\downarrow$ | 0.39 | $0.01 \pm 0.0001$ |
| | PSNR $\uparrow$ | 18.18 | $35.46 \pm 0.088$ |
| | MDE $\downarrow$ | 0.001 | $0.11 \pm 0.009$ |
| $\Omega_4$ | MSE $\downarrow$ | 0.99 | $0.01 \pm 0.0001$ |
| | PSNR $\uparrow$ | 18.14 | $38.45 \pm 0.052$ |
| | MDE $\downarrow$ | 0.02 | $0.12 \pm 0.011$ |
| $\Omega_5$ | MSE $\downarrow$ | 6.66 | $0.22 \pm 0.004$ |
| | PSNR $\uparrow$ | 15.39 | $30.17 \pm 0.082$ |
| | MDE $\downarrow$ | 0.40 | $0.08 \pm 0.016$ |

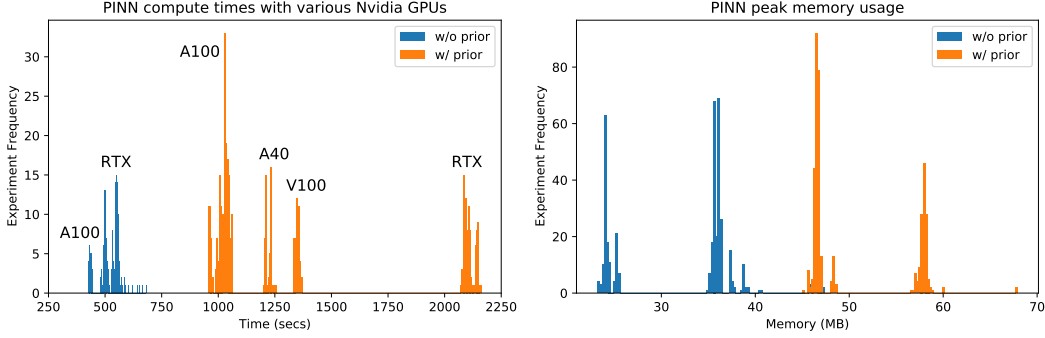

Figure 13: Histogram analysis of computation cost and memory usage for our proposed EB prior with PINN framework on 1000 random runs.

