# OpenReview forum: "Improved Training of Physics-Informed Neural Networks Using Energy-Based Priors: a Study on Electrical Impedance Tomography"
_ICLR.cc/2023/Conference — ICLR 2023 poster_

### Official Review · Reviewer_bAau · 2022-10-24

**Confidence:** 4
**Correctness:** 4
**Technical Novelty And Significance:** 4
**Empirical Novelty And Significance:** 4
**Recommendation:** 10

**Clarity, Quality, Novelty And Reproducibility:**

High quality paper trying to stuff a lot of information to small space that somewhat reduces the clarity. For example, one could bring the Full EIT Joint Training image to the actual text and move some of the text to the appendix. Just to get an earlier visual about the architecture in question. Novel solutions that solve essential stability problems.

**Strength And Weaknesses:**

This is an excellent paper. Introduction of the energy based criteria that quantifies the non-physical domain for the solutions is innovative and effective.

**Summary Of The Paper:**

The manuscript is using PINN to solve the problem of electrical inverse tomography. It provides examples where PINNs circumvent the instability of the EIT problem, it creates a data-driven energy-based model to improve the accuracy of the EIT reconstruction. The problem is not only mathematically interesting but has a significant practical use in multiple domains, not least in medicine.

Careful analysis is done on the different regularisation methods. These  show that viable solutions are found with regularisation. The traditional solutions are computed for comparison with different degrees of noise.

**Summary Of The Review:**

Excellent and informative paper.

---

> ### Author Response · Authors · 2022-11-16
> **Response to Reviewer bAau**
>
> #### Dear Reviewer bAau:
>
> We really appreciate your encouraging feedback.
>
> We have tried to bring the full-EIT setup into the main paper, but unfortunately, due to space constraints and given page limit, we were unable to organize it.

---

### Official Review · Reviewer_WoWz · 2022-10-24

**Confidence:** 3
**Correctness:** 3
**Technical Novelty And Significance:** 3
**Empirical Novelty And Significance:** 3
**Recommendation:** 6

**Clarity, Quality, Novelty And Reproducibility:**

The paper is well written> the use of EB: priors is novel for solving EIT semi-inverse problem, and the proposed technique seems to outperform the state-of-the-art. It seems that the authors are not intending to share their source code, which would hinder reproducibility.

**Strength And Weaknesses:**

Strengths:
1.	The proposed approach was demonstrated to be outperforming the strong solution proposed by Bar and Sochen, with faster convergence and more robustness toward the learning rate.
2.	The authors demonstrated the added value of EBM priors for different EIT approach, and also highlight the added value of added hinge loss.
3.
Weaknesses:

1.	The authors have only focused on the semi invers problem, whereas in real EIT experiments, one will only have access to current measurement on the body torso. Could the authors explain why they restricted themselves to the semi-inverse problem?
2.	Why did the authors not implement the classical FEM approach for solving the EIT problem, in order to compare their approach to a standard technique?
3.	Also, it would be interesting for the authors to report computational times and also memory usage. As this would help for comparison purpose.
4.	Could the authors also discuss how their proposed approach could extended to the 3D case?


**Summary Of The Paper:**

This paper introduces the use of Bayesian approach as prior for stabilizing and helping the training of Physics Inspired Neural Network (PINN) for solving the semi-inverse Electrical Impedance Tomography.

**Summary Of The Review:**

The authors present an interesting approach for stabilizing the training of PINN for EIT problems. The paper is well presented, and the results are convincing. It is a pity that the authors have only focused on the EIT semi inverse problem, and not the full inverse problem.

---

> ### Author Response · Authors · 2022-11-16
> **Response to Reviewer WoWz**
>
> #### Dear Reviewer WoWz,
> We really appreciate your insightful comments and feedback. We addressed your questions below. We also revised our paper accordingly (we have highlighted our revision).
>
>  ---
>
> ### "The authors have only focused on the semi-inverse problem, whereas in real EIT experiments, one will only have access to current measurement on the body torso. Could the authors explain why they restricted themselves to the semi-inverse problem?"
>
> This is a very valid point. Solving Full EIT using PINN is our main goal, but it is an extremely difficult problem. As also mentioned in the paper, we couldn’t successfully train full-EIT setup for the logan phantom. However, we have demonstrated the outcome of our model for simpler phantoms (Appendix A5). For these simpler phantoms, PINN w/o prior fails to learn any meaningful solution. We have left a general approach for the real-EIT setup for our future work.
>
> ---
>
> ### "Why did the authors not implement the classical FEM approach for solving the EIT problem, in order to compare their approach to a standard technique?"
>
> We have addressed this question with additional experiments using the FEM-based Levenberg-Marquard (LM) method. Our results for various phantoms are demonstrated in Appendix A.6 Fig. 11, 12. We compared the performance of our proposed full-EIT joint training with EB prior methodology and LM in Table 5. Our method outperforms the FEM baseline on all considered phantoms.
>
> Although comparing LM and semi-EIT is unfair as our semi-inverse problem utilizes internal domain data, we still report the results for the sake of completeness in Table 6.
>
> ---
>
> ### "Also, it would be interesting for the authors to report computational times and also memory usage. As this would help for comparison purposes."
>
> We report histogram analysis of computation time and memory usage in Appendix A.7 by running 500 random run experiments for vanilla PINN and another 500 random runs for PINN with EB prior. As expected, results show that PINN with EB prior uses slightly more resources, however the stability and accuracy guarantees clearly outweigh the meager increase in time and compute cost.
>
> ---
>
> ### "Could the authors also discuss how their proposed approach could be extended to the 3D case?"
>
> Application of EIT to 3D data is an actively studied area (Shin et al. 2020; Jehl et al. 2015; Ye et al. 2006). Our mathematical model will remain the same even in the case of 3D. However, there will be an additional spatial coordinate along the z-direction in the PINN input data. EBMs have also been successfully trained for point clouds (Xie et al. 2021), thus enabling one to extend our work to the 3D setup.
>
> * Xie J, Xu Y, Zheng Z, Zhu SC, Wu YN. Generative pointnet: Deep energy-based learning on unordered point sets for 3d generation, reconstruction and classification. InProceedings of the IEEE/CVF Conference on Computer Vision and Pattern Recognition 2021 (pp. 14976-14985).
>
> * Shin, Kwancheol, Sanwar Uddin Ahmad, and Jennifer L. Mueller. "A Three Dimensional Calderon-Based Method for EIT on the Cylindrical Geometry." IEEE Transactions on Biomedical Engineering 68.5 (2020): 1487-1495.
>
> * Jehl, Markus, et al. "Correcting electrode modelling errors in EIT on realistic 3D head models." Physiological Measurement 36.12 (2015): 2423.
>
> * Gang Ye, K. H. Lim, R. George, G. Ybarra, W. T. Joines and Q. H. Liu, "A 3D EIT system for breast cancer imaging," 3rd IEEE International Symposium on Biomedical Imaging: Nano to Macro, 2006., 2006, pp. 1092-1095, doi: 10.1109/ISBI.2006.1625112.
>
> ---
>
> ### “ It seems that the authors are not intending to share their source code, which would hinder reproducibility.”
>
> We will publish our code and configurations upon acceptance.

---

### Official Review · Reviewer_FzKV · 2022-10-30

**Confidence:** 3
**Correctness:** 3
**Technical Novelty And Significance:** 3
**Empirical Novelty And Significance:** 3
**Recommendation:** 8

**Clarity, Quality, Novelty And Reproducibility:**

The paper is well written and organized. It proposes an interesting idea incorporating PINN and DSM. No code is provided in the paper but there are some training and model details described in the appendix.

**Strength And Weaknesses:**

Strength:
- The paper is well written and organized.
- The studied problem and the proposed idea is interesting intersecting two areas of PINN and energy-based model, particularly denoising score matching (DSM)
- The results of simulation experiments and ablative study support the improvement of incorporating energy-based priors in the application of EIT

Weakness:
- How to train and obtain a EBM prior is an important and interesting part in the proposed method. For example, DSM always may require a large-scale dataset for training, but the paper “generates 6512 such smoothed phantoms for EBM prior training and 1628 for testing.” How can the author decide this is a sufficient dataset to train the EBM prior. Moreover, the distribution captured by EBM prior is always dependent on the training data. How could the trained EBM prior generalize to other cases or how easy is it to train an adaptive EBM prior for different application cases?
- The paper mentions that “ training of PINN is extremely sensitive to interplay between different loss terms and hyper-parameters, including the learning rate.” But for the proposed method, as shown in Eq. (11) as well as the training of EBM prior, there are still many hyper-parameters that need to be decided. Although Sec. 4.4 shows that with EBM prior, the model seems to have a higher training success rate, the training of EBM prior also depends on the different hyper-parameters choices and may also raise other problems during the training. How practical it could be to choose a reasonable set of hyper-parameters when generalized to different application cases?
- The paper conducts all the experiments on the simulation data. Is there any real dataset that could be used to validate the effectiveness of the proposed method in real applications? Especially, for the real data, is it still possible to get a decent scale of dataset to train the EBM prior and how efficient is it?
-For comparison among different baseline methods, beyond the quantitative results shown in Table 2, can the paper also provide the qualitative results comparison? It would be interesting how these methods may distinguish in the real application?
- The idea of combining PINN and DSM prior for solving inverse problems is interesting. The author introduces this mainly from the perspective of PINN and  solving the EIT inverse problem as in Sec. 5. There are also a series of researches recently for using DSM models for inverse problem solving such as medical image reconstruction by incorporating physics priors. How could this paper be relevant to those line of research also for inverse problem solving?
- In Table 2, the paper citations of titles are all missing?



**Summary Of The Paper:**

This paper proposes to incorporate energy-based priors to improve the training of physics-informed neural networks (PINN), for solving partial differential equation (PDE) based inverse problems. The method is mainly validated in the application of electrical impedance tomography (EIT).

**Summary Of The Review:**

The paper studies an interesting problem and proposes an interesting idea to incorporate energy-based priors to improve the training of physics-informed neural networks (PINN). But there may still be some concerns as aforementioned.

---

> ### Author Response · Authors · 2022-11-16
> **Response to Reviewer FzKV [2/2]**
>
> ### "Is there any real dataset that could be used to validate the effectiveness of the proposed method in real applications? Especially, for the real data, is it still possible to get a decent scale of dataset to train the EBM prior and how efficient is it? "
>
> Limitation of the current mathematical model in real-world data setting:
>
> 1. Our current mathematical model has access to internal measurements, contrary to the real-world setting, where we only have boundary measurements.
>
> 2. Although we have added enough noise to our simulated data, real-world data can be more unpredictable due to limitations in hardware, error in measurements, etc.
>
> 3. Real-world data collection is limited by the number of electrodes unlike in simulations, where we have access to all boundary and internal data.
>
> Possible remediations:
>
> 1. We can use any available heuristic or statistical approaches such as the works cited below, to first learn the interior $\sigma$ - distribution using the boundary data and then serve that reconstruction as an initialization for ML models.
>
> * Calvetti, D., S. Nakkireddy, and E. Somersalo. "Approximation of continuous EIT data from electrode measurements with Bayesian methods." Inverse Problems 35.4 (2019): 045012.
>
> * Nissinen, A., L. M. Heikkinen, and J. P. Kaipio. "The Bayesian approximation error approach for electrical impedance tomography—experimental results." Measurement Science and Technology 19.1 (2007): 015501.
>
> 2. We can make the simulated data as close as possible to the real-world setting by adding large amounts of noise and high shape deformity. The PINN will then try to learn to solve such complex data. In order to improve the informativeness of the EB prior itself, we can train the EBM on FEM reconstructions from various heuristic and statistical approaches instead of the current toy simulated data. These two solutions, when combined together, can be more effective at solving real-world EIT problems while keeping our base framework still intact.
>
> ---
>
> ### "No code is provided in the paper but there are some training and model details described in the appendix. "
>
> We will publish our code and configurations upon acceptance.

---

> ### Author Response · Authors · 2022-11-16
> **Response to Reviewer FzKV [1/2]**
>
> #### Dear Reviewer FzKV:
> We really appreciate your insightful comments and feedback. We addressed your questions below. We also revised our paper accordingly (we have highlighted our revision).
>
> ---
>
> ### “How can the author decide this is a sufficient dataset to train the EBM prior.”
> In general, the a-priori information for our Bayesian approach does not have to accurately model the output distribution but capture the high-level structure of the output to help the PINN avoid troublesome predictions such as predicting constant sigma for all domain points as shown in Fig 4. Here, any prior distribution that assigns a low probability to such prediction can help with the training of PINN.
>
> To better show this, we have extended our previous sensitivity analysis (Section 4.4) by adding a weak prior trained on 1% of our training data. In brief, we randomly sample 1% of data with replacement from the original training set (see samples in Appendix A.2 Figs.7) and train our EBM separately five times. We also repeat the EBM training with full data five times. For each of these EBMs, we randomly select 100 combinations of hyper-parameters and train the PINN with the prior. We reported the average number of successful experiments that reach a specific error threshold for each category of EB priors in Fig. 4.
>
> The results indicate that there is no considerable difference in the number of successful experiments. However, looking at the generated samples from each prior (Appendix A.4 Fig 8 and 9) we can infer that the EBMs trained with 1% of training data have lower quality compared to EBMs trained with all training data points (also confirmed by FID score), but still could learn some meaningful structures.
>
> ---
> ### “How could the trained EBM prior generalize to other cases or how easy is it to train an adaptive EBM prior for different application cases?”
>
> The prior must be trained on similar data points that share similar structures but does not require to be trained on exactly the same outputs. Particularly, Figs. 7 shows that the training data is different from our inference phantoms shown in Fig. 2.  We also included the nearest training data points (based on Euclidean distance) to each phantom in Fig 5. To better quantify this, the EB prior assigns energy levels in the range of [-30K, -24K] to training data and [-30K, -23K] to test data, while it assigns the energy level of -32K the Shepp-logan phantom (Phantom 1 in Fig 2).
>
> Our EBM prior PINN training framework is straightforward to extend to other PDEs. Users have the flexibility in choosing the mesh size to generate a set of discretized PDE solutions with various BCs for either the forward or inverse problem. The resolution of the mesh is directly proportional to the quality of the EBM prior and likewise the PINN results (high resolutions incur additional compute cost). Lastly, the PINN and EBM training procedures will remain exactly the same. Users have to first train their EBM separately on the above created set of discrete mappings. Then use PINNs with EB prior for combined training-inference to obtain solutions on unseen BCs.
>
> ---
>
> ### “How practical it could be to choose a reasonable set of hyper-parameters when generalized to different application cases?”
> Please note that EBM training is independent of PINNs, thus EBMs’ hyperparameters do not change the space of PINNs’ hyperparameters. We separately train the EBM, pick a good model and then search the space of PINNs’ hyperparameters.
>
> We choose the hyperparameters based on random search (Bergstra and Bengio, 2012) which is an effective approach when grid search is not practical due to the large space of hyperparameters as in our case. In the case of other applications, the number of loss terms and hyperparameters depends on the type of PDE and its problem-posedness. For instance, linear and well-posed problems don't need rigorous regularization and require fewer hyperparameters, and vice-versa for non-linear and ill-posed problems.
>
> Bergstra, J. and Bengio, Y. Random search for hyperparameter optimization. Journal of Machine Learning Research, 13:281–305, 2013.
>
> ---
>
> ### “There is also a series of research recently for using DSM models for inverse problem solving such as medical image reconstruction by incorporating physics priors. How could this paper be relevant to those lines of research also for inverse problem solving?”
>
> We have updated the related work Section 5 to extensively address this question. The important difference that set our work apart from other DSM prior works is that our application is to solve non-linear, ill-posed physics-based inverse problems of EIT. Most of the existing works rely on running inference over the posterior, which requires accessing the forward transformation (from reconstruction to the measurements). These techniques are not naively applicable to using PINNs for solving nonlinear PDEs such as the EIT problem.

---

### Author Response · Authors · 2022-11-16
**List of changes**

Dear Reviewers and AC,

We really appreciate your comments on our manuscript.

We have revised the paper with the following list of changes:

1. Section 2.1 - Updated mathematical notation for easier readability (Eq. 6,7,8 and 11).

2. Section 4.4 - Comparison of EBM priors of various strengths and normal PINN (Fig. 4).

3. Section 5 - Added relevant works regarding using prior in inverse problems.

4. Appendix A.3 - Training sample visualization (Fig. 5 and Fig. 7).

5. Appendix A.4 - New section for EBM prior sensitivity discussion (Fig. 8 and Fig. 9).

6. Appendix A.6 - New section for Finite Element Method (FEM) Baseline (Fig. 11, 12), (Table 5, 6).

8. Appendix A.7 - New section for Computation cost and memory usage analysis (Fig. 13).

All updated text has been highlighted in *red*. for your convenience.

We sincerely hope our response and revision address all the reviewers’ concerns.

Thank you very much.

Best regards,

Authors.

---

### Decision · Program_Chairs · 2023-01-20

**Decision:**

Accept: poster

**Justification For Why Not Higher Score:**

The method is evaluated in the limited context of EIT. Its utility beyond this domain is yet unknown.

**Justification For Why Not Lower Score:**

The method is clearly novel and well evaluated for the problem of EIT.

**Metareview: Summary, Strengths And Weaknesses:**

The paper develops an energy based model based regularizer in conjunction with a PINN to solve the ill-posed inverse problem of electrical inverse tomography. The method is shown to be faster than a classical PDE solution and more robust.

**Note From Pc:**

if the above contains the word "oral" or "spotlight" please see: "oral" presentation means -> notable-top-5% and "spotlight" means -> notable-top-25%. As stated in our emails, we are disassociating presentation type from AC recommendations